# Influence of Foliar Application of Hydrogen Peroxide on Gas Exchange, Photochemical Efficiency, and Growth of Soursop under Salt Stress

**DOI:** 10.3390/plants12030599

**Published:** 2023-01-29

**Authors:** Jessica Dayanne Capitulino, Geovani Soares de Lima, Carlos Alberto Vieira de Azevedo, André Alisson Rodrigues da Silva, Thiago Filipe de Lima Arruda, Lauriane Almeida dos Anjos Soares, Hans Raj Gheyi, Pedro Dantas Fernandes, Maria Sallydelândia Sobral de Farias, Francisco de Assis da Silva, Mirandy dos Santos Dias

**Affiliations:** 1Academic Unit of Agricultural Engineering, Federal University of Campina Grande, Campina Grande 58430-380, PB, Brazil; 2Academic Unit of Agrarian Sciences, Federal University of Campina Grande, Pombal 58840-000, PB, Brazil

**Keywords:** *Annona muricata* L., water salinity, attenuator, abiotic stress, reactive oxygen species

## Abstract

Hydrogen peroxide at low concentrations has been used as a salt stress attenuator because it induces a positive response in the antioxidant system of plants. This study aimed to assess the gas exchange, quantum yield, and development of soursop plants cv. Morada Nova grown with saline water irrigation and foliar hydrogen peroxide application. The experiment was carried out under greenhouse conditions using a randomized block design in a 4 × 4 factorial scheme corresponding to four levels of electrical conductivity of irrigation water, ECw (0.8, 1.6, 2.4, and 3.2 dS m^−1^), and four doses of hydrogen peroxide, H_2_O_2_ (0, 10, 20, and 30 μM), with three replicates. The use of irrigation water with electrical conductivity above 0.8 dS m^−1^ inhibited stomatal conductance, internal CO_2_ concentration, transpiration, maximum fluorescence, crown height, and vegetative vigor index of the Morada Nova cultivar of soursop. Compared to untreated plants, the hydrogen peroxide concentration of 30 μM resulted in greater stomatal conductance. Water salinity of 0.8 dS m^−1^ with hydrogen peroxide concentrations of 16 and 13 μM resulted in the highest variable fluorescence and quantum efficiency of photosystem II, respectively, of soursop plants cv. Morada Nova at 210 days after transplantation.

## 1. Introduction

The soursop (*Annona muricata* L.), a fruit tree native to Central America and the Peruvian Valleys, is notable for its commercialization potential in the domestic market with significant economic importance and export prospects because of the high acceptance of its fruit and pulp, primarily for numerous food and pharmaceutical applications [1,2]. Despite the production potential of soursop for the Northeast region, one of the limitations to its production system is the occurrence of low rainfall and high evapotranspiration in most months of the year [3]. In addition, the water sources commonly used in irrigation have high concentrations of salts [4,5].

High salt concentration in water and/or soil inhibits plant growth due to restrictions of water absorption (osmotic effect) and changes in metabolism, as well as ionic imbalance (specific ion effect), which affects photosynthetic pigments, harms cellular components, and causes lipid peroxidation of the membrane [6,7].

In soursop, salt stress induces loss of photosynthetic activity due to stomatal and non-stomatal limitations [8,9]. It compromises the quantum efficiency of photosystem II (PSII), demonstrating that PSII reaction centers have experienced photoinhibitory damage, which results in the formation of reactive oxygen species (ROS) [10] capable of inducing oxidative damage to proteins and other biological components [11].

Considering the socioeconomic importance of soursop in Brazilian agribusiness, it is essential to study strategies that enable the use of saline water in its cultivation in semi-arid regions. Among the alternatives used to reduce the effects of salt stress on plants, the application of hydrogen peroxide (H_2_O_2_) on leaves stands out [12]. Hydrogen peroxide is a reactive oxygen species (ROS) that, when applied in low concentrations, acts in acclimatization and/or signaling to salt stress due to metabolic alterations that are responsible for increasing its tolerance to stress, thus enabling its use of water with high concentrations of salts [13,14,15]. It must be considered that the beneficial effect of the application of H_2_O_2_ depends on several factors, including the concentration, plant species analyzed, development stage, and application method [16].

In recent years, studies have reported that foliar application with hydrogen peroxide can attenuate the deleterious effects caused by salt stress in several crops, for example, cotton [12], passion fruit [13], mini-watermelon [17], orange [18], tomato [19], maize [20], wheat [16], and rice [21]. However, information about its use in fruit plants is incipient, especially in the soursop crop.

This study is based on the hypothesis that the soursop crop suffers smaller losses from salt stress under the application of hydrogen peroxide through the regulation of physiological processes, which contributes to an increase in photosynthetic and antioxidant activity, avoiding lipid peroxidation caused by ROS and an increase in the rate of CO_2_ assimilation and stomatal regulation, thus leading to an improvement in the growth of the soursop plant. In this context, this study aimed to evaluate the gas exchange, quantum yield, and growth of soursop cv. Morada Nova irrigated with saline water under the foliar application of hydrogen peroxide.

## 2. Results

### 2.1. Leaf Gas Exchange

The salinity of irrigation water had a significant effect on the soursop plants’ stomatal conductance (*gs*), internal CO_2_ concentration (*Ci*), and transpiration € (Table 1). H_2_O_2_ concentrations, as a single factor, influenced stomatal conductance (*gs*). The CO_2_ assimilation rate (*A*), instantaneous water use efficiency (*WUEi*), and instantaneous carboxylation efficiency (*CEi*) were all significantly affected by the interaction between water salinity levels and H_2_O_2_ concentrations.

The stomatal conductance of soursop plants cv. Morada Nova was linearly reduced by the increase in the electrical conductivity of irrigation water (Figure 1A), with a decrease of 18.08% per unit increment in ECw. When comparing the *gs* of plants subjected to ECw of 3.2 dS m^−1^ to that of plants that received the lowest salinity level (0.8 dS m^−1^), there was a reduction of 50.74%.

Regarding the effects of H_2_O_2_ concentrations on the stomatal conductance of soursop plants (Figure 1B), there was a linear increase corresponding to 5.25% per unit increment in H_2_O_2_ concentration. In relative terms, the foliar application of 30 μM resulted in an increase of 0.0870 H_2_O m^−2^ s^−1^ compared to plants grown under 0 μM of hydrogen peroxide.

Regarding the effects of the electrical conductivity of irrigation water on the internal CO_2_ concentration (Figure 1C) and transpiration (Figure 1D), there were linear reductions of 6.72 and 9.47%, respectively, per unit increase in ECw. When comparing the *Ci* and *E* of plants grown under water salinity of 3.2 dS m^−1^ to the values of those that received ECw of 0.8 dS m^−1^, there were reductions of 17.04 and 24.59%, respectively, at 210 days after transplantation.

For the interaction between the ECw levels and H_2_O_2_ concentrations (Figure 1E), the maximum estimated value of the CO_2_ assimilation rate (3.20 μmol CO_2_ m^−2^ s^−1^) was obtained in plants irrigated with an ECw of 0.80 dS m^−1^ and in the absence of H_2_O_2_ application. On the other hand, the minimum value of *A* (0.05 μmol CO_2_ m^−2^ s^−1^) was reached in plants subjected to ECw of 3.2 dS m^−1^ and the foliar application of hydrogen peroxide at a concentration of 30 μM.

For instantaneous carboxylation efficiency (*CEi*) (Figure 2A) and instantaneous water use efficiency (*WUEi*) (Figure 2B), irrigation with water of electrical conductivity of 0.8 dS m^−1^ in the absence of foliar application of hydrogen peroxide (0 μM) resulted in higher estimated values of CEi (0.0278 (μmol m^−2^ s^−1^) (μmol mol^−1^)^−1^) and *WUEi* (2.67 (μmol CO_2_ m^−2^ s^−1^) (mmol H_2_O m^−2^ s^−1^)^−1^). On the other hand, irrigation with an ECw of 3.2 dS m^−1^ and the foliar application of H_2_O_2_ at concentrations of 27 and 30 μM, respectively, led to the estimated minimum values of 0.008 ((μmol m^−2^ s^−1^) (μmol mol^−1^)^−1^) in *CEi* and 1.46 ((μmol m^−2^ s^−1^) (mmol H_2_O m^−2^ s^−1^)^−1^) in *WUEi*.

### 2.2. Quantum Yield

The initial fluorescence, variable fluorescence, and quantum efficiency of photosystem II of soursop plants at 210 DAT were significantly affected by the interaction between water salinity levels and hydrogen peroxide concentrations (Table 2). The salinity levels of the irrigation water significantly influenced the maximum fluorescence of the soursop. Hydrogen peroxide concentrations, as a single factor, influenced the initial and variable fluorescence and the quantum efficiency of photosystem II of soursop plants.

The interaction between water salinity levels and hydrogen peroxide concentrations significantly influenced the initial fluorescence of soursop (Figure 3A), with the highest estimated value (674.52) attained in plants irrigated with ECw of 0.8 dS m^−1^ without the foliar application of H_2_O_2_. Plants under irrigation with water of 3.2 dS m^−1^ and foliar application of 30 μM of H_2_O_2_ reached the estimated minimum value of 552.10.

The maximum fluorescence of soursop plants was linearly reduced by the increase in the electrical conductivity of irrigation water (Figure 3B), with a decrease of 2.85% per unit increment in ECw. When comparing the Fm of plants irrigated with an ECw of 3.2 dS m^−1^ to that of plants under irrigation with the lowest level of water salinity (0.8 dS m^−1^), a decrease of 6.99% was verified.

Irrigation with water of 0.8 dS m^−1^ associated with the application of hydrogen peroxide concentrations of 16 and 13 μM, respectively, promoted the highest estimated values of variable fluorescence (1875) and quantum efficiency of photosystem II (0.746). However, the combination between the highest salinity of irrigation water (3.2 dS m^−1^) and the highest concentration of H_2_O_2_ (30 μM) resulted in the lowest estimated values of *Fv* (1592.37) and *Fv*/*Fm* (0.690).

### 2.3. Morphological Parameters

The interaction between irrigation water salinity levels and hydrogen peroxide concentrations significantly affected stem diameter (SD) and crown diameter (D_crown_) (Table 3). The single factor of irrigation water salinity influenced the crown height (H_crown_) and the vegetative vigor index (VVI) of soursop plants 210 days after transplantation. There was a significant effect of hydrogen peroxide concentrations only on the V_crown_ of soursop plants.

The salinity of irrigation water negatively affected the crown height (Figure 4A) and vegetative vigor index (Figure 4B) of soursop plants at 210 days after transplantation. Regression equations (Figure 4A,B) showed linear reductions of 3.32 and 4.23% per unit increment in ECw in the H_crown_ and VVI of soursop plants, respectively. When comparing the H_crown_ and VVI of plants irrigated with water of an electrical conductivity of 3.2 dS m^−1^ to the values of those subjected to water salinity of 0.8 dS m^−1^, there were reductions of 8.20 and 10.51%, respectively.

The interaction between water salinity levels and hydrogen peroxide concentrations significantly affected the crown volume (Figure 4C), crown diameter (Figure 4D), and stem diameter (Figure 4E) of soursop plants. Irrigation with water of 0.8 dS m^−1^ in the absence of the foliar application of H_2_O_2_ promoted the maximum values of 0.125 m^3^ and 1.47 m, respectively, for V_crown_ and D_crown_. However, water salinity levels of 0.8 and 3.2 dS m^−1^ under the foliar application of 30 μM of H_2_O_2_ contributed to the minimum values of V_crown_ (0.1238 m^3^) and D_crown_ (0.762 m), respectively, in soursop plants cv. Morada Nova.

Regarding stem diameter (Figure 4E), it was verified that plants receiving water of 0.8 dS m^−1^ and exposed to a 30 μM H_2_O_2_ concentration achieved a stem diameter (SD) of 21.11 mm. On the other hand, water salinity of 3.2 dS m^−1^ associated with the foliar application of 11 μM of H_2_O_2_ resulted in a lower value of the stem diameter (17.96 mm).

## 3. Discussion

The excess of salts present in irrigation water induces salt stress in plants, which negatively affects their metabolism and limits their growth and development, standing out as a limiting factor for irrigated agriculture, especially in semi-arid regions [22]. In the present study, it was verified that the increase in the electrical conductivity of the irrigation water reduced the gas exchanges, the quantum yield, and the growth of the soursop plant, but the deleterious effects caused by the saline stress were partially mitigated by the foliar application of hydrogen peroxide.

The reduction of *gs* (Figure 1A) with increasing water salinity is a way for plants to minimize water losses in the form of vapor to the atmosphere and maintain turgor pressure inside their cells, in addition to reducing the absorption of salts [23,24]. Reductions in stomatal conductance due to salt stress were also observed in studies with other fruit plants such as acerola [24], guava [25], and custard apple [26]. The beneficial effect of H_2_O_2_ observed on the plants’ stomatal conductance (Figure 1B) may have occurred due to the defense mechanisms of the plant, inducing the system of antioxidant enzymes, thus minimizing the harmful effects of salinity [27,28]. Silva et al. [15], in their study evaluating the induction of salt stress tolerance (ECw ranging from 0.6 to 3.0 dS m^−1^) in soursop seedlings using hydrogen peroxide (from 0 to 20 μM), found that the application of hydrogen peroxide at a concentration of 20 μM promoted greater stomatal conductance compared to the control treatment (0 μM H_2_O_2_) for all salinity levels at 110 days after transplantation.

Increasing the electrical conductivity of irrigation water also reduced the transpiration of soursop plants (Figure 1D). A decrease in *E* is a strategy of plants to reduce water loss through transpiration, constituting a mechanism of tolerance to salt stress [29].

The decrease in *A* (Figure 1E) may be related to the lower concentrations of CO_2_ found in the substomatal chamber due to the partial closure of the stomata and possible metabolic restrictions to the Calvin cycle and, consequently, a diminution in the synthesis of sugars in the photosynthetic process and in the substrate for RuBisCo [30,31]. The stress-induced decrease in CO_2_ assimilation rate in plants can be caused by stomatal and/or non-stomatal factors, leading to changes in the metabolic processes of photosynthesis and affecting the activities of a number of enzymes in the stroma involved in CO_2_ reduction [32,33].

Salt stress in plants causes significant losses in the functioning of the photosystem due to the degradation of the proteins involved in the photosynthetic activity. This may explain the reductions in *CEi* (Figure 2A) and *WUEi* (Figure 2B) in soursop plants cv. Morada Nova with the increase in the electrical conductivity of irrigation water [34]. Silva et al. [35] also observed a decrease in the carboxylation efficiency in passion fruit under salt stress (electrical conductivity of water varying from 0.7 to 2.8 dS m^−1^) and application of hydrogen peroxide (0, 25, 50, and 75 μM) at 60 days after transplantation.

In the present study, it was verified that the foliar application of hydrogen peroxide at a concentration of 30 µM intensified the effects of salt stress on the initial fluorescence (Figure 3A). In this case, the concentration of 30 μM may have induced oxidative damage to the cell membrane and possibly had a negative influence on the initial fluorescence of soursop plants cv. Morada Nova at 210 DAT. At high concentrations, H_2_O_2_ causes damage to plants, presumably because of alterations in their metabolism, mainly as a consequence of oxidative stress, which limits photosynthetic activities [36].

The restriction of maximum fluorescence (Figure 3B) by salt stress indicates a slowdown in photosynthetic activity aimed at mitigating the toxic effects of salinity [37]. In a study conducted by Silva et al. [38] evaluating the fluorescence of chlorophyll *a* in soursop plants under saline stress (ECw ranging from 0.8 to 4.0 dS m^−1^), a reduction of 3.31% in maximum fluorescence was also observed by an increase in the electrical conductivity of the irrigation water; the authors attributed this fact to the low efficiency in the photoreduction of quinones and in the flow of electrons between the photosystems, which results in the low activity of photosystem II in the thylakoid membrane, directly influencing the flow of electrons between the photosystems.

The variable fluorescence (Figure 3C) and quantum efficiency of photosystem II (Figure 3D) benefited from the application of hydrogen peroxide at estimated concentrations of 16 and 13 μM, respectively. Thus, it can be concluded that the use of hydrogen peroxide at low concentrations contributed to greater efficiency in the photoreduction of quinone A and the flow of electrons between the photosystems, promoting the adequate activity of PSII in the membrane of thylakoids, directly influencing the flow of electrons between the photosystems, which indicates that PS II was not damaged because, when the photosynthetic apparatus is intact, the values of *Fv*/*Fm* vary between 0.75 and 0.85 [39]. This result is similar to that reported by Veloso et al. [40], who evaluated the photochemical efficiency and growth of soursop rootstocks subjected to salt stress (ECw ranging from 0.6 to 3.0 dS m^−1^) and hydrogen peroxide (0 and 20 μM) and found that applications of hydrogen peroxide at the concentration of 20 μM minimized the negative effects of salinity on the initial fluorescence and favored the variable fluorescence and quantum efficiency of photosystem II at 120 days after sowing.

The results obtained in the present study indicate that the growth of soursop plants is negatively affected by the increase in the salinity of the irrigation water. Inhibition of plant growth may be a consequence of the effect caused by excess salts in the root zone, which imposes water limitations, negatively affecting cell elongation and expansion. In addition, the partial closure of stomata compromises photosynthesis, resulting in lower growth [3,41].

On the other hand, the foliar application of hydrogen peroxide at an estimated concentration of 11 µM promoted an increase in growth in stem diameter. Veloso et al. [42], while evaluating the physiological changes and growth of soursop cultivated under saline waters and H_2_O_2_ in the post-grafting stage, reported that the exogenous application of H_2_O_2_ at 20 μM reduced the harmful effect of water salinity on the stem diameter of the rootstock and scion of soursop plants irrigated with water of 1.6 dS m^−1^ at 150 days after transplantation.

In general, the results of this research reveal that the salt stress caused by the irrigation water up to 3.2 dS m^−1^ negatively affected the gas exchanges, the photochemical efficiency, and the growth of the soursop plants under the conditions of a protected environment. These alterations may be related to osmotic and ionic effects, particularly of Na^+^ and Cl^−^, which interfere with the metabolic processes, causing damage to the membrane, nutritional imbalance, changes in the levels of growth regulators, and decreases in the synthesis of chlorophyll [43,44]. However, the foliar application of hydrogen peroxide in low concentrations can reduce the harmful effects of the salinity of irrigation water on the soursop plant. This fact may be related to enzymatic antioxidant defense mechanisms (catalase and peroxidase) in plants, reducing the negative effect of reactive oxygen species [45,46]. Additionally, hydrogen peroxide can improve the absorption of water and nutrients, including elements such as N, P, and K that are essential for plant growth and development [47].

On the other hand, at higher concentrations of hydrogen peroxide, an effect was observed on the analyzed variables. It is important to note that the beneficial effect of hydrogen peroxide depends on several variables, including the concentration of the solution; i.e., at higher concentrations, H_2_O_2_ can exert hazardous effects on plants [12,48]. Hydrogen peroxide is the most stable reactive oxygen species in cells and, at high concentrations, can quickly spread across the subcellular membrane, resulting in oxidative damage to the plasma membrane [16]. Additionally, at high concentrations, hydrogen peroxide can react with O_2_ and possibly become responsible for the dissociation of the pigment-protein complex of the internal antenna of the PS II light-gathering system within the photosynthetic apparatus, leading to enzymatic inactivation, pigment discoloration, and lipid peroxidation [45,49].

## 4. Materials and Methods

### 4.1. Location of the Experiment

The experiment was carried out between April and November 2020 under the conditions of the greenhouse that belongs to the Academic Unit of Agricultural Engineering of the Federal University of Campina Grande, situated in the municipality of Campina Grande, Paraíba, Brazil, at the geographic coordinates of 07°15′18″ S, 35°52′28″ W with a mean altitude of 550 m. Figure 5 shows the temperature (maximum and minimum) and average relative air humidity data for the experimental site.

### 4.2. Treatments and Experimental Design

The experimental design was randomized in a 4 × 4 factorial arrangement. The treatments consisted of the combination of two factors: four levels of electrical conductivity of irrigation water (ECw; 0.8, 1.6, 2.4, and 3.2 dS m^−1^) associated with four concentrations of hydrogen peroxide (H_2_O_2_; 0, 10, 20, and 30 μM), with three replicates and one plant per plot, totaling forty-eight experimental units.

The salinity levels of the water were established based on a study conducted by [15], who observed that water up to 2.0 dS m^−1^ can be used to produce soursop seedlings with an acceptable average reduction (up to 10%) in growth. Concentrations of hydrogen peroxide were based on the results of an assay conducted by [9], who verified that the use of hydrogen peroxide at the concentration of 20 μM attenuated the harmful effects of irrigation water salinity on the initial growth and gas exchange of soursop cv. Morada Nova.

### 4.3. Description of the Experiment

Plastic recipients adapted as drainage lysimeters with 200 L capacity were used to grow the plants, and each lysimeter was drilled at the base to allow the drainage of excess water and connected to a transparent drain of 16 mm diameter. The end of the drain inside the lysimeter was wrapped with a nonwoven geotextile (Bidim OP 30) to prevent clogging by soil material. A container was placed below each drain to collect drained water and determine water consumption by the plants (Figure 6).

The pots were filled with a soil classified as *Entisol* with sandy loam texture from the rural area (0–0.30 m layer) of the municipality of Riachão de Bacamarte, PB, Brazil, whose chemical and physical characteristics (Table 4) were obtained according to the methodologies recommended by [50].

The saline waters were obtained by the addition of sodium chloride, calcium chloride, and magnesium chloride salts in the equivalent proportion of 7:2:1, a predominant ratio found in the principal sources of water in northeastern Brazil [51], following the relationship between electrical conductivity and the salt concentration [52], according to Equation (1):(1)
Q = 640 × ECw
where:

Q = Quantity of salts to be applied (mg L^−1^);

ECw = Electrical conductivity of water (dS m^−1^).

After transplanting the seedlings to the lysimeters, irrigation was performed manually and applied daily to each container at 5 p.m., with the volume corresponding to that obtained by the water balance, according to Equation (2):(2)$$VI=\frac{\left(Va-Vd\right)}{\left(1-LF\right)}$$
where:

VI = Volume of water to be applied in the irrigation event (mL);

Va = Volume applied in the previous irrigation event (mL);

Vd = Volume drained (mL);

LF = Leaching fraction of 0.10.

The soursop seedlings cv. Morada Nova were acquired from a commercial nursery accredited in the Registry of Seeds and Seedlings, located in the District of São Gonçalo, Sousa-PB, species registered under No. 23458 in the national registry of cultivars (RNC) of the Ministry of Agriculture, Livestock and Supply of Brazil, and were produced in polyethylene bags with dimensions of 10 × 20 cm. Mineral fertilization was performed according to the recommendations of [53], applying 40 g of N, 60 g of K_2_O, and 40 g of P_2_O_5_ per plant per year. Urea, potassium chloride, and monoammonium phosphate (MAP) were used as sources of nitrogen, potassium, and phosphorus, respectively.

The fertilizer doses were split into 24 portions and applied every 15 days. Micronutrients were applied from 60 days after transplantation (DAT) and continued at fortnightly intervals with Dripsol micro solution (2.5 g L^−1^) with the following composition: N (15%), P_2_O_5_ (15%), K_2_O (15%), Ca (1%), Mg (1.4%), S (2.7%), Zn (0.5%), B (0.05%), Fe (0.5%), Mn (0.05%), Cu (0.5%), and Mo (0.02%), by spraying on the adaxial and abaxial sides of the leaves.

The different concentrations of hydrogen peroxide were obtained by dilution in distilled water, followed by calibration in a spectrophotometer at an absorbance wavelength of 240 nm. Foliar applications started at 30 DAT of the seedlings to the lysimeters and were performed at 30-day intervals, spraying the abaxial and adaxial sides of the leaves to obtain complete wetting using a backpack sprayer between the hours of 17:00 and 18:00. On average, 330 mL of H_2_O_2_ solution was applied per plant in each application. The air drift between treatments was controlled by a plastic tarpaulin curtain which involved the entire plant as the hydrogen peroxide solution was applied (Figure 7).

Formative pruning was carried out as the plant reached 60 cm height when the apical meristem bud was cut. Of the shoots that emerged, three well-distributed and equidistant branches were selected, and these branches, in turn, were pruned when they reached 40 cm in length [54]. During the experimental period, the emergence of pests and diseases was monitored by observing their incidence, and they were eradicated by chemical control using recommended insecticides/pesticides.

### 4.4. Variables Analyzed

Gas exchange, chlorophyll *a* fluorescence, and the growth of soursop cv. Morada Nova were evaluated at 210 DAT. Gas exchange was evaluated by CO_2_ assimilation rate (*A*) (μmol CO_2_ m^−2^ s^−1^), transpiration (*E*) (mmol H_2_O m^−2^ s^−1^), stomatal conductance (*gs*) (mol H_2_O m^−2^ s^−1^), and internal CO_2_ concentration (*Ci*) (μmol CO_2_ m^−2^ s^−1^) in the leaves of the middle third of the plants using a portable IRGA (infrared gas analyser, LCpro-SD model, ADC BioScientific, UK). The ratios *A/gs* and *A/Ci* were utilized to obtain *WUEi*, the water use efficiency ((μmol CO_2_ m^−2^ s^−1^) (mmol H_2_O m^−2^ s^−1^)^−1^), and *CEi*, the carboxylation efficiency ((μmol m^−2^ s^−1^) (μmol mol^−1^)^−1^), respectively. Observations were taken between 07:00 and 10:00 a.m. on the third fully expanded leaf counted from the apical bud under natural conditions of air temperature, CO_2_ concentration, and employing an artificial source of radiation established through the photosynthetic response curve to light and determination of the point of photosynthetic saturation by light [55].

Chlorophyll *a* fluorescence measurements were made on the same leaves utilizing a pulse-modulated fluorometer, OS5p model from Opti Science. Initial fluorescence (*F*_0_), maximum fluorescence (*Fm*), variable fluorescence (*Fv*), and the quantum efficiency of photosystem II (*Fv/Fm*) were measured; this protocol was performed after adaptation of the leaves to the dark for 30 minutes between 06:00 and 09:00 a.m., utilizing a clip of the device to ensure that all the primary acceptors were fully oxidized.

The growth of soursop was evaluated by measuring crown height (H_crown_), stem diameter (SD), and crown diameter (D_crown_), which was considered the average crown diameter in the row direction (RD) and in the inter-row direction (IRD). Crown volume (V_crown_) and the vegetative vigor index (VVI) were determined by Equations (3) and (4), respectively, following the methodology of [56]:(3)$$V_{\mathrm{crown}\;}=\frac{\pi \;\text{×}\;H\;\text{×}\;\mathrm{RD}\;\text{×}\;\mathrm{IRD}}{6}$$
(4)$$\mathrm{VVI}=\frac{\left[H\;+\;D_{\mathrm{crown}}+\left(\mathrm{SD}+10\right)\right]}{100}$$
where:

V_crown_—Crown volume (m^3^);

VVI—Vegetative vigor index;

H—Crown height (m);

RD—Crown diameter in the row direction (m);

IRD—Crown diameter in the inter-row direction (m);

SD—Stem diameter (m).

### 4.5. Statistical Analysis

The collected data were subjected to the distribution normality test (Shapiro–Wilk test) at a 0.05 probability level. Subsequently, analysis of variance was performed at a 0.05 and 0.01 probability level, and in the cases of significance, a regression analysis was performed using the statistical program SISVAR-ESAL [57]. The choice of model was based on the coefficient of determination. In the case of the significance of the interaction between factors, TableCurve 3D software was used to create the response surfaces.

## 5. Conclusions

Physiological indices of soursop are negatively affected by increases in the electrical conductivity of irrigation water above 0.8 dS m^−1^. However, the foliar application of hydrogen peroxide between concentrations of 10 and 30 µM mitigates the harmful effects of salinity on the gas exchange, quantum yield, and growth in stem diameter of soursop cv. Morada Nova at 210 days after transplantation. These results reinforce the hypothesis that the foliar application of hydrogen peroxide in adequate concentrations can act as a signaling molecule, influential in mitigating salt stress in soursop plants, which can potentialize the use of brackish water in irrigated agriculture, mainly in regions with a scarcity of fresh water. However, further studies are necessary to understand how hydrogen peroxide acts in salt stress signaling through biochemical analysis. In addition, it is fundamental to perform research under field conditions to prove the beneficial effects of hydrogen peroxide in the attenuation of salt stress in soursop plants.

## Figures and Tables

**Figure 1 plants-12-00599-f001:**
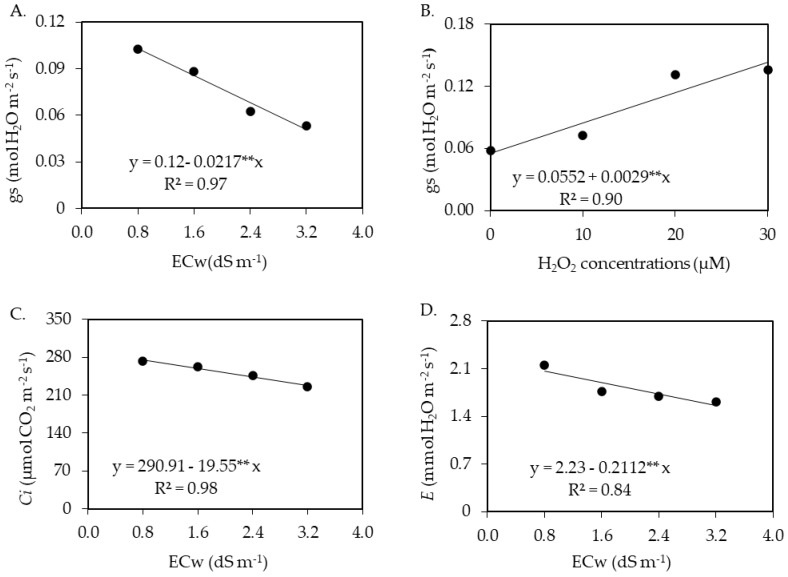
Stomatal conductance (*gs*) of soursop plants cv. Morada Nova as a function of the salinity of irrigation water (ECw) (**A**) and concentrations of hydrogen peroxide (H_2_O_2_) (**B**), internal CO_2_ concentration (*Ci*) (**C**), and transpiration (*E*) (**D**) as a function of ECw and CO_2_ assimilation rate (*A*) (**E**) as a function of the interaction between ECw levels and H_2_O_2_ at 210 days after transplantation. ** is, significant at *p* ≤ 0.01 by the F-test. x and y correspond to ECw and hydrogen peroxide (H_2_O_2_) concentrations, respectively.

**Figure 2 plants-12-00599-f002:**
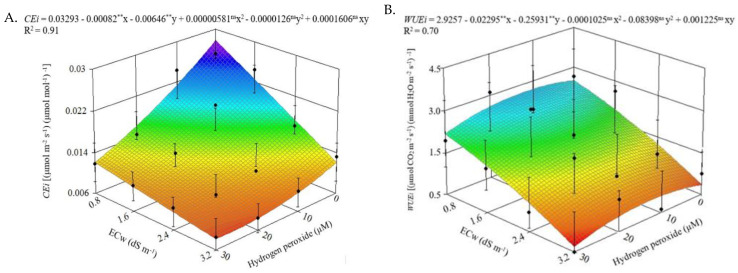
Instantaneous carboxylation efficiency (*CEi*) (**A**) and instantaneous water use efficiency (*WUEi*) (**B**) of soursop plants cv. Morada Nova as a function of the interaction between the irrigation water salinity levels (ECw) and concentrations of hydrogen peroxide (H_2_O_2_) at 210 days after transplantation. ns and ** are, respectively, not significant and significant at *p* ≤ 0.01 by the F-test. x and y correspond to ECw and hydrogen peroxide (H_2_O_2_) concentrations, respectively.

**Figure 3 plants-12-00599-f003:**
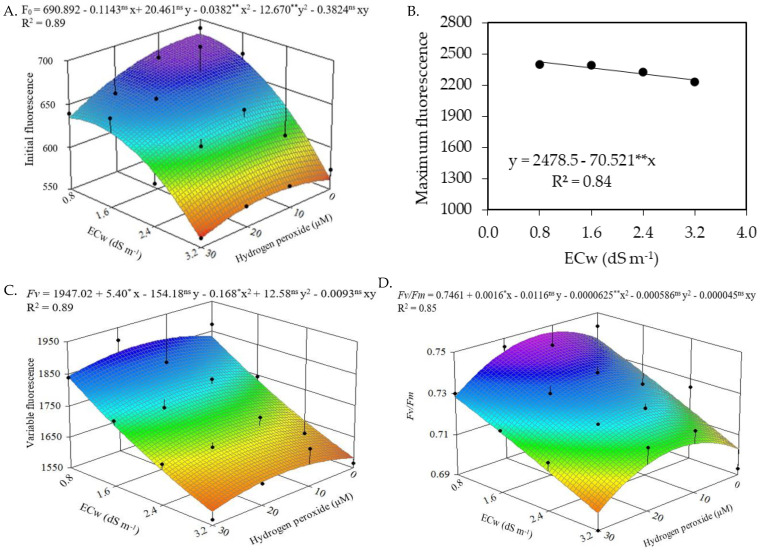
Initial fluorescence (*F*0) (**A**), variable fluorescence (*Fv*) (**C**), and quantum efficiency of photosystem II (*Fv*/*Fm*) (**D**) of soursop plants cv. Morada Nova as a function of the interaction between the salinity levels of irrigation water (ECw) and hydrogen peroxide concentrations, and maximum fluorescence (*Fm*) (**B**) as a function of the irrigation water salinity (ECw) at 210 days after transplantation. ns, * and ** are, respectively, not significant and significant at *p* ≤ 0.05 and ≤ 0.01 by the F-test. x and y correspond to ECw and hydrogen peroxide (H_2_O_2_) concentrations, respectively.

**Figure 4 plants-12-00599-f004:**
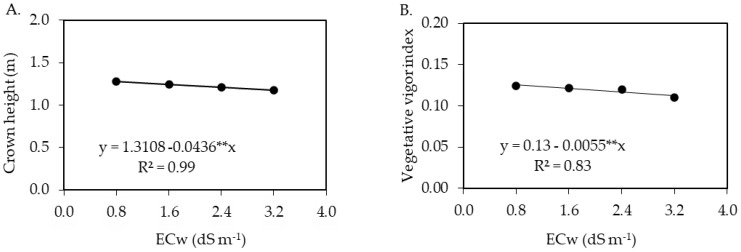
Crown height (Hcrown) (**A**) and vegetative vigor index (VVI) (**B**) of soursop plants cv. Morada Nova as a function of the irrigation water salinity (ECw), and crown volume (Vcrown) (**C**), crown diameter (Dcrown) (**D**), and stem diameter (SD) (**E**), as a function of the interaction between the salinity of irrigation water (ECw) and hydrogen peroxide concentrations at 210 days after transplantation. ns and ** are, respectively, not significant and significant at *p* ≤ 0.01 by the F-test. x and y correspond to ECw and hydrogen peroxide (H_2_O_2_) concentrations, respectively.

**Figure 5 plants-12-00599-f005:**
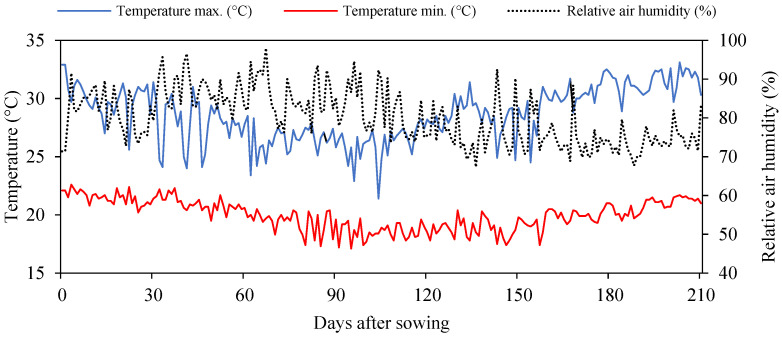
Observed daily temperature (maximum and minimum) and average relative humidity of air in the internal area of the greenhouse during the experimental period.

**Figure 6 plants-12-00599-f006:**
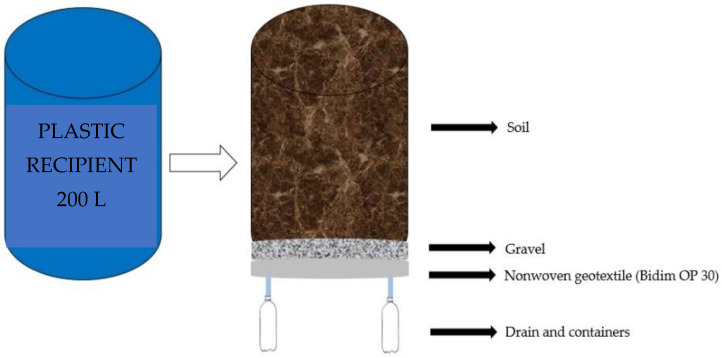
Illustration of filling drainage lysimeters.

**Figure 7 plants-12-00599-f007:**
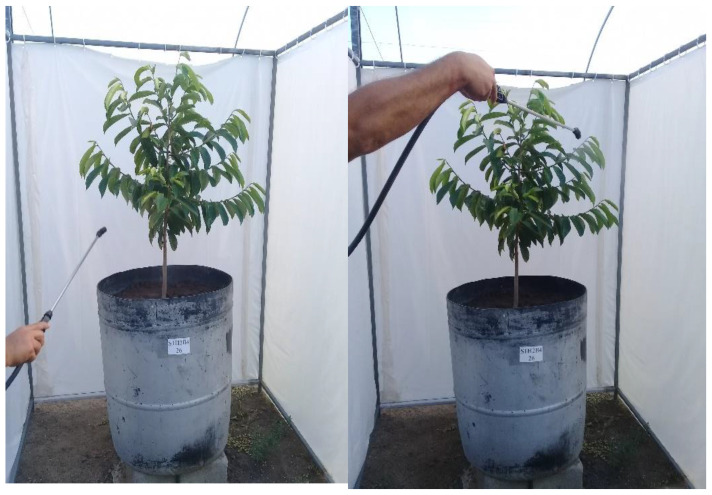
Application of hydrogen peroxide on the abaxial and adaxial sides of soursop leaves.

**Table 1 plants-12-00599-t001:** Analysis of variance (F-test) summary of stomatal conductance (*gs*), transpirati€(*E*), CO_2_ assimilation rate (*A*), internal CO_2_ concentration (*Ci*), instantaneous carboxylation efficiency (*CEi*), and instantaneous water use efficiency (*WUEi*) of soursop plants cv. Morada Nova irrigated with saline water and subjected to foliar application of hydrogen peroxide at 210 days after transplantation.

Source of Variation	DF	Mean Squares					
		*gs*	*E*	*A*	*Ci*	*CEi*	*WUEi*
Salinity levels (SL)	4	0.000547 **	0.1781 **	0.880 ^ns^	87.729 **	0.00009 ^ns^	0.3614 ^ns^
Linear regression	1	0.000240 *	0.111 **	-	122.90 **	-	-
Quadratic regression	1	0.00010 ^ns^	0.1220 ^ns^	-	124.32 ^ns^	-	-
Hydrogen peroxide (H_2_O_2_)	4	0.001836 **	0.0206 ^ns^	2.650 *	3594.42 ^ns^	0.000083 **	3.1369 **
Linear regression	1	0.004160 **	-	0.0811 ^ns^	-	0.00009 ^ns^	5.693 **
Quadratic regression	1	0.000300 ^ns^	-	4.656 **	-	0.00030 **	0.0965 ^ns^
Interaction (SL × H_2_O_2_)	16	0.001775 ^ns^	0.3500 ^ns^	9.450 **	2442.85 ^ns^	0.000186 **	1.074 **
Blocks	3	0.00015 ^ns^	0.0687 ^ns^	0.1692 ^ns^	23.003 ^ns^	0.000053 ^ns^	0.2020 ^ns^
Residual	30	0.000149	0.0347	0.334	105.27	0.000060	0.112
CV (%)		15.65	10.33	12.39	3.97	10.26	13.91

^ns^, * and ** are, respectively, not significant and significant at *p* ≤ 0.05 and *p* ≤ 0.01. CV: coefficient of variation; DF: degrees of freedom.

**Table 2 plants-12-00599-t002:** Analysis of variance (F-test) summary of initial fluorescence (*F*_0_), maximum fluorescence (*Fm*), variable fluorescence (*Fv*), and quantum efficiency of photosystem II (*Fv/Fm*) of soursop plants cv. Morada Nova irrigated with saline water and subjected to foliar application of hydrogen peroxide at 210 days after transplantation.

Source of Variation	DF	Mean Squares			
		*F* _0_	*Fm*	*Fv*	*Fv/Fm*
Salinity levels (SL)	4	7842.51 **	75277.36 **	139489.6 **	0.002239 **
Linear regression	1	19634.8 **	190976.0 **	415168.01 **	0.006202 **
Quadratic regression	1	2257.7 ^ns^	32870.5 ^ns^	3088.02 ^ns^	0.000033 ^ns^
Hydrogen peroxide (H_2_O_2_)	4	658.99 **	1771.91 ^ns^	4881.95 *	0.000928 **
Linear regression	1	436.32 *	-	608.01 ^ns^	0.000375 ^ns^
Quadratic regression	1	21.60 ^ns^	-	13736.3 **	0.002408 **
Interaction (SL × H_2_O_2_)	16	362.87 **	7207.74 ^ns^	5538.60 **	0.000163 *
Blocks	3	140.15 ^ns^	16949.8 ^ns^	1220.47 ^ns^	0.000065 ^ns^
Residual	30	82.83	3186.54	1279.87	0.000049
CV (%)		1.36	2.42	2.08	0.97

^ns^, * and ** are, respectively, not significant and significant at *p* ≤ 0.05 and ≤ 0.01. DF: degrees of freedom; CV: coefficient of variation.

**Table 3 plants-12-00599-t003:** Analysis of variance (F-test) summary of stem diameter (SD), crown height (H_crown_), crown diameter (D_crown_), crown volume (V_crown_), and vegetative vigor index (VVI) of soursop plants cv. Morada Nova irrigated with saline water and subjected to foliar application of hydrogen peroxide at 210 days after transplantation.

Source of Variation	DF	Mean Squares				
		SD	H_crown_	D_crown_	V_crown_	VVI
Salinity level (SL)	4	0.00002 ^ns^	0.0250 *	0.0748 ^ns^	0.00262 *	0.0007 *
Linear regression	1	-	0.0297 *	-	0.00662 _**_	0.00001 *
Quadratic regression	1	-	0.0285 ^ns^	-	0.00063 ^ns^	0.00002 ^ns^
Hydrogen peroxide (H_2_O_2_)	4	0.000021 ^ns^	0.0129 ^ns^	0.0344 ^ns^	0.01284 **	0.00006 ^ns^
Linear regression	1	-	-	-	0.01460 **	-
Quadratic regression	1	-	-	-	0.0107 ^ns^	-
Interaction (SL × H_2_O_2_)	16	0.000021 **	0.0109 ^ns^	0.1119 **	0.0067 ^ns^	0.00005 ^ns^
Blocks	3	0.000030 ^ns^	0.0520 ^ns^	0.0230 ^ns^	0.00129 ^ns^	0.000075 ^ns^
Residual	30	0.000023	0.078	0.0201	0.0045	0.00004
CV (%)		7.29	7.28	15.81	15.39	1.63

^ns^, * and ** are, respectively, not significant and significant at *p* ≤ 0.05 and ≤ 0.01. CV: coefficient of variation; DF: degrees of freedom.

**Table 4 plants-12-00599-t004:** Chemical and physical characteristics of the soil (0–0.30 m depth) used in the experiment.

Chemical characteristics								
pH_H2O_	OM	P	K^+^	Na^+^	Ca^2+^	Mg^2+^	Al^3+^	H^+^
1:2.5	g dm^−3^	mg dm^−3^	cmol_c_ kg^−1^					
6.5	8.1	79	0.24	0.51	14.9	5.4	0	0.9
Chemical characteristics				Physical characteristics				
EC_se_	CEC	SAR_se_	ESP	Particle-size fraction (g kg^−1^)			Moisture (dag kg^−1^)	
dS m^−1^	cmol_c_ kg^−1^	(mmol L^−1^)^0.5^	%	Sand	Silt	Clay	33.42 kPa ^1^	1519.5 kPa ^2^
2.15	16.54	0.16	3.08	572.7	100.7	326.6	25.91	12.96

pH: hydrogen potential, OM: Organic matter, Walkley–Black Wet Digestion; Ca^2+^ and Mg^2+^ extracted with 1 M potassium chloride at pH 7.0; Na^+^ and K^+^ extracted with 1 M ammonium acetate at pH 7.0; Al^3+^ + H^+^ extracted with 0.5 M calcium acetate at pH 7.0; ECse: electrical conductivity of saturated paste extract; CEC: cation exchange capacity; SARse: sodium adsorption ratio of saturated paste extract; ESP: exchangeable sodium percentage; superscripts 1 and 2 correspond to field capacity and permanent wilting point, respectively.

## Data Availability

Not applicable.

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
