# Peer review of "Influence of Foliar Application of Hydrogen Peroxide on Gas Exchange, Photochemical Efficiency, and Growth of Soursop under Salt Stress"

_plants, 2023, doi:10.3390/plants12030599_

Round 1

Reviewer 1 Report

The paper titled : “Influence of Foliar Application of Hydrogen Peroxide on Gas Exchange, Photochemical Efficiency, and Growth of Soursop Under Salt Stress”, submitted by the authors Capitulino et al investigated the some physiological parameters including gas exchange and quantum yield as well as morphological parameters  including crown height and vegetative vigor of soursop cv. Morada Nova, irrigated with saline water under foliar application of different concentrations of hydrogen peroxide.

The paper contains good amount of data about the selected topic (use of stress alleviating materials) and is if special interest for researchers within this field.   

There are some things need to be addressed before the publishing of this paper:

1.       In the abstract:

-          Lines 28-29, add “compared to untreated plants” to the sentence “Hydrogen peroxide concentration of 30 μM resulted in greater stomatal conductance ” and under saline conditions of …

-          Line 29 , replace “higher” with the highest.

2.       The introduction part need some improvement and organization. Try to rewrite it into four distinct paragraphs and not as present form (scattered topics). The last paragraph include the objective of the work.

3.       In the results and discussion section,  

The section is hard to follow its subtopics, so you need to divide it into numbered subsections to be able to follow each parameter studied.

4.       In the materials and methods part

-          In the section please  specify the person who identified the cultivar used in the study and the voucher number of specimen.

5.       The conclusion: Please add the prospect of the work  

I give you major revision.

Author Response

Campina Grande, PB

Jan. 17, 2023

Reference: Plants 2165352 – Response to Review Report I

Dear Editor

The authors are very grateful to you and the Reviewers for the positive and constructive comments and suggestions on our manuscript entitled “Influence of Foliar Application of Hydrogen Peroxide on Gas Exchange, Photochemical Efficiency, and Growth of Soursop Under Salt Stress”. The authors would like to inform you that a thorough revision of the manuscript was made, incorporating the suggestions and adapting the text according to the comments. Attached is the revised version of the manuscript. All changes in the text are highlighted in red color.

The authors remain at your disposal for any further information and explanation.

The responses/clarifications to the issues raised by the Reviewer 1/Editor are presented below:

REVIEWER 1

Comments and Suggestions for Authors

There are some things need to be addressed before the publishing of this paper:

  1. IN THE ABSTRACT:

- Lines 28-29, add “compared to untreated plants” to the sentence “Hydrogen peroxide concentration of 30 μM resulted in greater stomatal conductance” and under saline conditions of …

Response:  The suggestion was accepted and inserted in the revised version of the manuscript, as can be seen on page 1, line 26.

- Line 29, replace “higher” with the highest.

Response:  The word higher was replaced as suggested by the reviewer.

  1. INTRODUCTION:

- The introduction part need some improvement and organization. Try to rewrite it into four distinct paragraphs and not as present form (scattered topics). The last paragraph include the objective of the work.  

Response: The introduction has been redrafted following reviewers' suggestions.

  1. RESULTS AND DISCUSSION:

 - The section is hard to follow its subtopics, so you need to divide it into numbered subsections to be able to follow each parameter studied

Response: As suggested by the reviewers, the Results and Discussion section was separated and divided into subtopics in the revised version of the manuscript.

  1. MATERIALS AND METHODS:

-  In the section, please specify the person who identified the cultivar used in the study and the voucher number of specimen.

Response: The species was registered (n° 23458) by the Brazilian Ministry of Agriculture, Livestock and Supply, this information was inserted in the revised version of the manuscript, as can be seen on page 13, between lines 376 and 377.

  1. conclusion:

Please add the prospect of the work

Response: The conclusion was reformulated in the revised version of the manuscript, taking into consideration  the suggestions of the reviewer.

Yours sincerely,

Geovani Soares de Lima

Reviewer 2 Report

The article investigated gas exchange, quantum yield and development of soursop cultivar Morada Nova grown with salt water irrigation and foliar application of hydrogen peroxide.

There are several questions:

Increase the number of keywords.

The introduction should be expanded to include more references to recent research.

In my opinion, the results and the discussion should be separated.

In conclusion, describe directions for further research.

There are doubts about the originality of the article, since the authors have already performed similar studies before: https://doi.org/10.14295/cs.v10i4.3036

http://dx.doi.org/10.5433/1679-0359.2020v41n6Supl2p3023

The novelty of the conducted research should be indicated. Why did the authors decide to repeat the work if the main results were published earlier?

Author Response

Campina Grande, PB

Jan. 17, 2023

Reference: Plants 2165352 – Response to Review Report II

Dear Editor

The authors are very grateful to you and the Reviewers for the positive and constructive comments and suggestions on our manuscript entitled “Influence of Foliar Application of Hydrogen Peroxide on Gas Exchange, Photochemical Efficiency, and Growth of Soursop Under Salt Stress”. The authors would like to inform you that a thorough revision of the manuscript was made, incorporating the suggestions and adapting the text according to the comments. Attached is the revised version of the manuscript. All changes in the text are highlighted in red color.

The authors remain at your disposal for any further information and explanation.

The responses/clarifications to the issues raised by the Reviewer 2/Editor are presented below:

REVIEWER 2

Reviewer’s comments to Authors

THERE ARE SEVERAL QUESTIONS:

  1. Increase the number of keywords

Response: The number of keywords has been increased, as suggested.

  1. The introduction should be expanded to include more references to recent research.

Response:  The suggestion was accepted and the introduction has been expanded as suggested by the reviewers.

  1. In my opinion, the results and the discussion should be separated.

Response:  As suggested, the item Results and Discussion in the revised version of the manuscript has been separated.

  1. In conclusion, describe directions for further research.

Response: The item conclusion was reformulated in the revised version of the manuscript, taking into consideration the reviewers' suggestions.

  1. There are doubts about the originality of the article, since the authors have already performed similar studies before:

https://doi.org/10.14295/cs.v10i4.3036

http://dx.doi.org/10.5433/1679-0359.2020v41n6Supl2p3023

The novelty of the conducted research should be indicated. Why did the authors decide to repeat the work if the main results were published earlier?

Response: In the article (https://doi.org/10.14295/cs.v10i4.3036) gas exchange and soursop growth under salt stress and hydrogen peroxide application were analyzed. However, this study was carried out during the seedling formation phase and the treatment with hydrogen peroxide was carried out by soaking the seeds and later hydrogen peroxide  was also applied to the leaves, which differs completely from the present study and the paper submitted to Plants.

 In the article http://dx.doi.org/10.5433/1679-0359.2020v41n6Supl2p3023, changes in gas exchange, chloroplastic pigments, and cell damage in soursop plants irrigated with saline water and under foliar application of hydrogen peroxide in the post-grafting phase were evaluated, i.e., the plants used in this study were grafted. Furthermore, only one concentration of hydrogen peroxide (20 µM) was tested.In the present study, gas exchange, quantum yield, and growth parameters not addressed in previous studies were evaluated. It is noteworthy that in this study concentrations (0, 10, 20, and 30 µM) of hydrogen peroxide were tested for a longer period of application (210 DAT), which makes the study different from the previous one.

Yours sincerely,

Geovani Soares de Lima

Round 2

Reviewer 1 Report

Accepted for me 

Reviewer 2 Report

The authors have corrected all comments. I have no more questions.